# Garlic (*Allium sativum* L.) as an Ally in the Treatment of Inflammatory Bowel Diseases

Silvana Zugaro, Elisabetta Benedetti * and Giulia Caioni

Department of Life, Health and Environmental Sciences, University of L'Aquila, 67100 L'Aquila, Italy
* Correspondence: elisabetta.benedetti@univaq.it

**Abstract:** For centuries, garlic (*Allium sativum*) has been used both as a traditional remedy for most health-related ailments and for culinary purposes. Current preclinical investigations have suggested that dietary garlic intake has beneficial health effects, such as antioxidant, anti-inflammatory, antitumor, antiobesity, antidiabetic, antiallergic, cardioprotective, and hepatoprotective effects. Its therapeutic potential is influenced by the methods of use, preparation, and extraction. Of particular importance is the Aged Garlic Extract (AGE). During the aging process, the odorous, sour, and irritating compounds in fresh raw garlic, such as allicin, are naturally converted into stable and safe compounds that have significantly greater therapeutic effects than fresh garlic. In AGE, S-allylcysteine (SAC) and S-allylmercaptocysteine (SAMC) are the major water-soluble organosulfurized compounds (OSCs). SAC has been extensively studied, demonstrating remarkable antioxidant, anti-inflammatory, and immunomodulatory capacities. Recently, AGE has been suggested as a promising candidate for the maintenance of immune system homeostasis through modulation of cytokine secretion, promotion of phagocytosis, and activation of macrophages. Since immune dysfunction plays an important role in the development and progress of various diseases, given the therapeutic effects of AGE, it can be thought of exploiting its immunoregulatory capacity to contribute to the treatment and prevention of chronic inflammatory bowel diseases (IBD).

**Keywords:** *Allium sativum*; aged garlic extract; allicin; organosulfur compounds; cytokines; inflammatory bowel disease; diseases; health benefits

## 1. Introduction

Garlic (*Allium sativum*) is one of the oldest cultivated plants. It has been used as a spice, food, and folklore medicine for over 4000 years [1]. In ancient Egypt and Rome, garlic was given to laborers and soldiers, perhaps to mitigate fatigue or to aid in healing from physical exhaustion [2]. In ancient Greece, it was consumed to treat intestinal and lung ailments [3]. Louis Pasteur first reported the antibacterial properties of garlic as early as 1858 [4], although, in India, garlic has been used for centuries as an antiseptic lotion to wash wounds and ulcers. In China, garlic tea has long been recommended for fever, headache, cholera, and dysentery. In Japan, garlic-containing miso soup is used as a remedy for the common cold with headaches, fever, and sore throat [5].

More than 3000 publications in this century have provided evidence of garlic's effectiveness in preventing and treating a variety of diseases, validating the traditional uses.

A wide range of therapeutic effects of garlic has been observed, including anticancer, antibacterial, antiviral, antidiabetic, antihypertensive, cardioprotective, hepatoprotective, and hypolipidemic effects [6]. In addition to the pharmacological activities mentioned above, garlic is a possible modifier of various biological responses, which has led many scientists to focus on the anti-inflammatory, antioxidant, and immunostimulant properties exerted by the bioactive compounds of garlic [7]. Considering that some diseases can be caused by immune dysfunctions and oxidative stress, which is often associated with a state of acute or chronic inflammation, garlic's modification of immune functions (and thus of inflammatory and oxidative stress) may contribute to disease treatment and prevention [8].

### 1.1. Garlic's Bioactive Components

Garlic contains more than 200 chemical compounds; it is composed of acids, 1.5% of fibers, 1.2% free amino, 2% proteins, 2.3% organosulfurized compounds (OSC), 28% carbohydrates, and 65% water. It also contains fat-soluble vitamins (E, K, and A) and water-soluble vitamins (C and B complex), and minerals (Na, Mg, Fe, K, P, Ca, and Zn) (Figure 1) [9,10]. The OSCs, responsible for the pungent odor and spicy flavor, are among the major contributors to its pharmacological properties [11]. The existence and potency of the bioactive constituents of garlic vary based on its method of preparation and extraction [12].

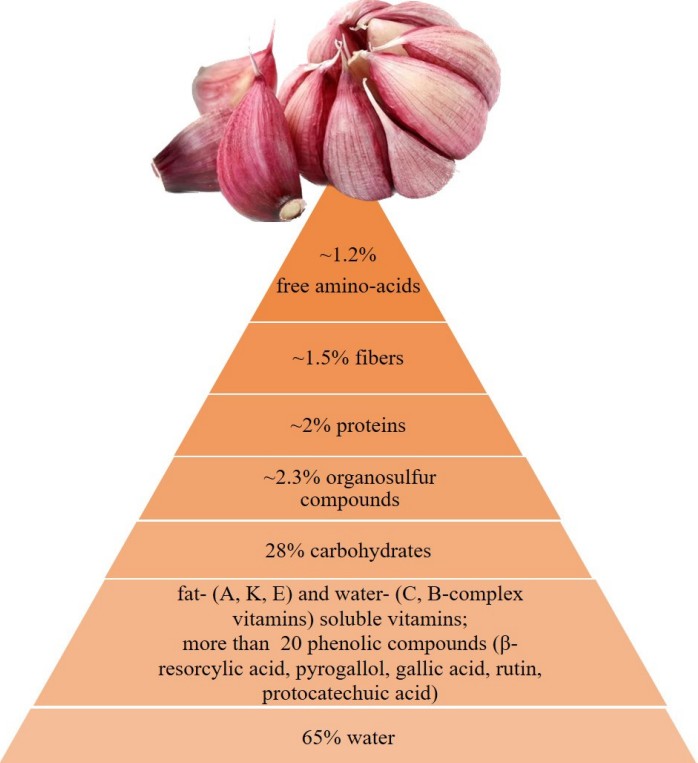

**Figure 1.** Fresh garlic composition and phenolic compounds.

The main sulfur-containing compounds in garlic are S-allyl-L-cysteine sulfoxide (Alliin) and γ-glutamyl-S-allyl-L-cysteine (GSAC). Allicin, derived from alliin, is considered the main bioactive compound present in the aqueous extract or the homogenate of raw garlic [9,10]. Alliin is a natural compound found in intact, raw garlic that forms complexes with lliinase, a vacuolar enzyme with liasic activity that is released by crushing, cutting, and grinding the garlic bulb. Sulfenic acid, pyruvic acid, and ammonia are among the reactive intermediates produced by the lliinase reaction. Sulfenic acid, an unstable chemical molecule, undergoes self-condensation to produce allicin, a sulfur compound that is soluble in fat [8].

Since its discovery, around the 1940s, various reports have emerged regarding the beneficial biological activities of allicin such as its antimicrobial, antioxidant, and anti-inflammatory effects, suppression of cholesterol biosynthesis, and antitumor activity [10]. However, unfavorable chemical properties of allicin, such as high reactivity and instability, have also been detected, raising doubts about its real effectiveness and beneficial characteristics [11,13]. It decomposes instantly into other compounds, such as diallyl sulfide (DAS), diallyl disulfide (DADS), diallyl trisulfide (DAT), ditins, and ajoene (Figure 2) [8,11]. Allicin has shown its beneficial properties in vitro, while its effects in vivo are uncertain: the allinase is inactivated around a pH 3.5 or lower, which is commonly found in the stomach and leads to a low release of allicin. Additionally, it was observed that after the addition of allicin to the circulation, it could not be detected in the blood, due to its rapid metabolism

(half-life < 1 min). Moreover, after the ingestion of raw garlic, allicin or its by-products could not be traced in urine or stool. This enzyme is also thermosensitive [11–13].

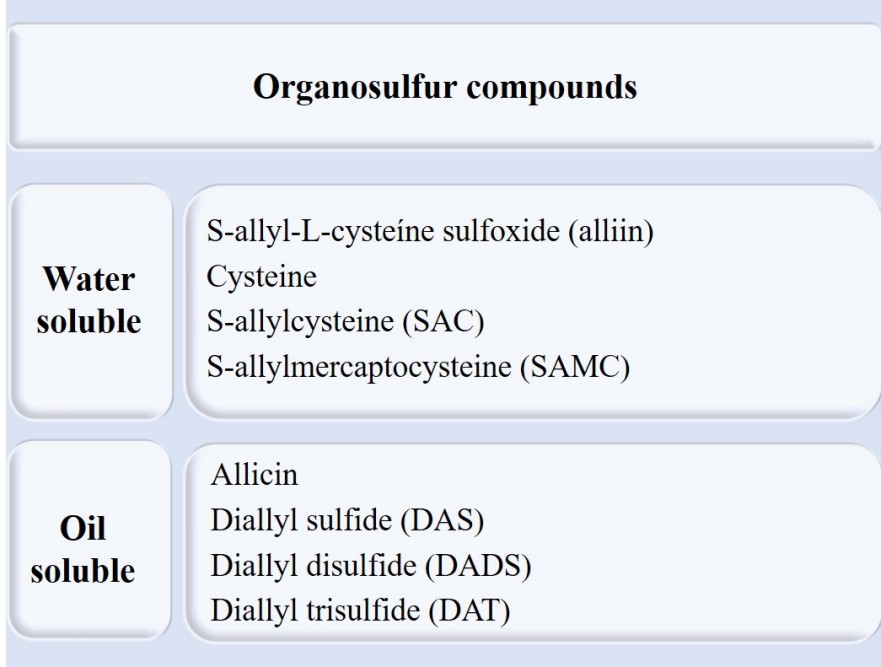

**Figure 2.** Organosulfur compounds of garlic.

Powdered garlic is a clove of garlic simply dehydrated and pulverized, and its composition, along with its allinase activity, is identical to fresh garlic. However, to avoid inactivating the allinase, the dehydration temperature should not be higher than 60 °C [14,15].

Oil macerates were originally used as condiments, and they consist of encapsulated blends of whole garlic cloves ground in vegetable oil. During the manufacturing process, part of the alliin is converted into allicin. Since allicin is unstable and decomposes rapidly, oily macerate preparations contain compounds decomposed by allicin such as ditins, ajoene and sulfides, residual amounts of alliin, and other constituents of garlic [16].

Although garlic is one of the best-known medicinal plants for its effects on cancer prevention, and immunomodulatory and antioxidant properties, unfortunately, garlic is also associated with numerous adverse effects including odor, allergic reactions (allergic contact dermatitis, generalized urticaria, angioedema, pemphigus, and anaphylactic reaction), as well as photo-allergies, gastrointestinal disturbances and the hypocoagulant effect [11,13].

The detrimental effects of garlic's compounds may occur when they are ingested in high amounts or when an individual has a sensitivity or allergy to them. In several papers, weight loss and the onset of anemia were found in mice after being subjected to prolonged feeding of high levels of raw garlic [17]. In other studies, in addition to the clastogenic capacity of raw garlic juice, it caused the death of rats from stomach injuries at a dose of 5 mL/kg [17,18]. The surviving rats showed swelling of the liver, hypertrophy of the spleen and adrenal glands, and decreased erythrocyte counts with various morphological changes after 3 and 8 days. A 10-day treatment showed significantly higher levels of aspartate aminotransferase (AST) due to liver injury [19]. As demonstrated by Harunobu et al. (2001), raw garlic juice can create significant damage to the epithelial mucous membrane after only 2 h in rats. Moreover, ulcers, narrowing, and bleeding in the epithelial mucosa were detected after 24 h of exposure [13]. In addition, chronic administration of garlic powder resulted in the inhibition of spermatogenesis in rats and significant liver injury, while oil of garlic given in high doses after 24 h of fasting was fatal for causing acute pulmonary edema with severe congestion [19].

## 1.2. Beneficial Effects of Aged Garlic Extract

A novel garlic preparation known as Aged Garlic Extract (AGE) was created in an attempt to lessen these characteristics without compromising the biological properties. It is obtained from the aging of fresh garlic in aqueous ethanol (15–20%) for more than 10 months at room temperature and has no strong odor or sour and irritating taste [13,19,20]. During the aging process, the odorous, sour, and irritating compounds in fresh raw garlic, such as allicin, are naturally converted into stable and safe compounds [13]. These include the presence of phenols, flavonoids, pyruvate, thiosulfate, S-allylcysteine (SAC) and S-allylmercaptocysteine (SAMC), allixin, and selenium, which are highly bioavailable and antioxidants. Another compound present in AGE is N-alpha-(1-deoxy-D-fructos-1-yl)-L-arginine (Fru-Arg), whose activity is comparable to ascorbic acid [13,14].

In AGE, SAC and SAMC are the main water-soluble OSCs and the SAC has been extensively studied demonstrating the possession of various biological activities such as anticancer, antioxidant, hypocholesterolemic, antihepatotoxic, and neuroprotective effects [13]. The safety of AGE has been confirmed by numerous toxicity tests from which it emerged that SAC is 30 times less toxic than other typical garlic compounds such as allicin and diallyldisulfide (DADS) [13]. The 50% lethal oral dose (LD50) of SAC in female (9.39 g/kg) or male (8.89 g/kg) mice is higher than both allicin (female: 0.363 g/kg and male: 0.309 g/kg) and diallyldisulfide (female: 1.3 g/kg and male: 0.145 g/kg), while negative effects are observed only after oral administration of SAC at high doses ($\geq$500 mg/kg) for a prolonged period of administration (about a month) [13]. Unlike fresh garlic extract, AGE is characterized by the high bioavailability of its active ingredients, in particular by SAC [21]. In an experimental study, the half-life of the SAC was determined after its oral administration to volunteers; the half-life was greater than 10 h, while the elimination time was greater than 30 h [13,21]. In addition, it is readily absorbed in the gastrointestinal tract and can be detected in different tissues up to 8 h after dosing in particular. It is mainly distributed to the plasma, liver, and kidney [21].

Many favorable biological and pharmacological effects of the consumption of garlic preparations have been reported experimentally and clinically. Garlic has been shown to reduce risk factors for cardiovascular disease, i.e., lowering serum cholesterol and triglycerides, inhibiting blood clotting, improving blood circulation, and lowering blood pressure [22]. In a Canadian study, consumption of 7.2 g of garlic extract per day for six months showed a 5.5% reduction in systolic blood pressure, a 7.0% reduction in total serum cholesterol, and a reduction in 4.6% of the low-density lipoprotein cholesterol compared to placebo given to 41 moderately hypercholesterolemic men aged 32 to 68 years [23]. Additionally, in studies of rats fed a cholesterol-fed diet, supplementing with garlic extract reduced triacylglycerols and cholesterol compared to controls [24].

The ancient Egyptians used garlic externally for the treatment of tumors [4], as did the doctors of ancient India. Current research has confirmed the medicinal benefits of garlic constituents in the prevention or treatment of cancer, including organo-sulfur compounds deriving from AGE [25]. In particular, there is compelling evidence that the consumption of certain vegetables, such as garlic, reduces the risk of colorectal, stomach, lung, esophagus, breast and bladder cancers by modulating many pathways including alteration of the enzymes that metabolize the carcinogen, the arrest of the cell cycle and the induction of apoptosis and the suppression of the oncogenic signaling pathways [26,27]. The inhibition of neoplastic transformation appears to be greater, especially by the OSCs deriving from AGE. SAMC's ability to remove ROS and over-regulate antioxidant enzymes may partially explain its anti-cancer properties [28,29]. SAMC induces the family of Bcl-2 genes to activate the apoptosis of tumor cells through the modulation of the MAPK pathway and the release of the mitochondrial cytochrome c [6]. At the same time, it inhibits the proliferation of malignant cells by inducing histone acetylation and inhibiting the polymerization of microtubules. It was also found that AGE causes a decrease in the chemo-resistance of cancer cells by down-regulation of Bcl-2 and up-regulation of E-cadherin, which leads to a decrease in tumor invasiveness [6]. Therefore, garlic compounds could significantly

influence tumor development, complementing their antiproliferative action and their anti-inflammatory/immunostimulating properties [6]. The main properties of bioactive compounds present in AGE are reported in Table 1.

**Table 1.** Main properties of bioactive compounds in Aged Garlic Extract.

| Properties | Examples |
|---|---|
| Antimicrobial | *Bacillus cereus, Staphylococcu aureus, Escherichia coli, Klebsiella pneumoniae, Proteus mirabilis., Aspergillus versicolor, Penicillum citrinum, Penicillium expansum.* |
| Antifungal | *Candida tropicalis, Blastoschizomyes capitatus, Trichoderma harzianum, Candidia albicans, Botrytis cinerea.* |
| Hematological and Cardiovascular effects | Anti-thrombotic effects, anticoagulant, antiplatelet, reduction of triglycerides, LDL, and increased HDL. |
| Antioxidant | Increased expression of antioxidant enzymes (SOD, catalase and GSH), Scavenge radicals |
| Anti-cancer | Anti-inflammatory, anti-angiogenic apoptosis induction: Bcl-2↓, Bax↑, P53↑, activation of JNK1. |
| Anti-inflammatory | Inhibition of NF-κB, reducing the inflammatory response mediated by TNF-α ↓, IL-1↓, IL-6↓, MCP-1↓ and IL-12↓. |

Anti-oxidant, Anti-Inflammatory/Immuno-Stimulating Properties of Aged Garlic Extract

The effects of AGE compounds on human health were investigated mainly in terms of anti-oxidant properties and immunomodulation, and this information has significant practical implications, particularly for diseases such as inflammatory bowel disease (IBD) that are characterized by significant inflammatory processes. The antioxidant properties of SAC are probably due to the presence of a thiol group capable of donating its proton to an electrophilic species, neutralizing it, or making it less reactive [30]. SAC is known to eliminate the superoxide anion [31], the hydrogen peroxide ($H_2O_2$) [30,31], the hydroxyl radical (● OH) [28,32], and the peroxynitrite anion ($ONOO^-$) as well as hypochlorous acid (HOCl) and singlet oxygen ($^1O_2$) [24,32]. As reported by Takeda et al. (1981), chronic oral administration of AGE-enriched food (2% *w/w*) for 2 months to mice with premature senescence increased survival rates, reduced memory acquisition deficits, averted decreased brain weight and atrophy of the frontal region of the brain, and reduced skin roughness and hair loss [33]. Assays performed on in vitro systems, consisting of endothelial cells and macrophages, revealed the role of AGE in protecting cells from oxidized-LDL injury [34]. AGE and SAC were reported to induce anti-oxidant enzymes, such as NAD(P)H: quinone oxidoreductase, and Nrf2 factor [20]. Thus, allowing an enhancement of the endogenous cellular antioxidant defenses, such as the increase in the enzymatic levels of superoxide dismutase (SOD), which dismantles superoxide, and of glutathione peroxidase, which destroys toxic peroxides and small molecules including glutathione [35].

The reduction of oxidative stress AGE-mediated has important implications in the protection of gastric tissue during inflammatory events, such as ulcers. For example, a study performed on an experimental rat model of indomethacin-induced ulcer highlighted the gastro-protective properties of AGE and, therefore, its ability to resolve histopathological abnormalities in gastric tissue by reducing oxidative stress and increasing the gastric level of prostaglandin E2, glutathione and nitric oxide [36]. The gastroprotective power may occur also acting as an inhibitor of tumorigenesis. S-allylmercapto-L-cysteine (SAMC) was shown to slow down the growth rate of gastric cancer cells in vitro and increase their death. These results were consistent with SAMC involvement in modulating the expression levels of Bcl-2 and Bax [37].

Another way that AGE exerts its anti-oxidant properties is by inhibiting pro-oxidant enzymes. SAC can inhibit the production of nitric oxide synthase (NOS) through the suppression of iNOS genes in macrophages and hepatocytes [38]. The administration of SAC and AGE was observed to decrease the abundance of NADPH oxidase catalytic subunit [39]. Cyclooxygenase (COX) catalyzes the formation of prostaglandins, thromboxane, and levuloglandins. The COX-1 isoform is expressed in many tissues with functions including the protection of the gastric mucosa and the regulation of vascular tone [40]. In a model of cerebral ischemia, it was found that the administration of AGE (1.2 mL/kg ip., onset of reperfusion) exerted a neuroprotective effect attributable to its ability to decrease the increase induced by ischemia of 8-hydroxy-2-deoxyguanosine (a marker of oxidative DNA damage) and TNF-$\alpha$ (a marker of inflammation) [41]. AGE can also modulate COX-2 activity. This enzyme is involved in inflammatory processes, being induced with different stimuli such as LPS, interleukin-1 (IL-1), IL-2, and tumor necrosis factor (TNF-$\alpha$), it also produces $O_2\bullet^-$ from $O_2$ generating oxidative stress [40].

AGE was shown to have immunomodulatory effects in different studies. According to certain research, aged garlic extract may assist to enhance the efficiency of some immune cells that are essential for the immunological response, including T-cells and natural killer cells [42]. Regarding the immune-stimulating properties, garlic fructans deserve particular attention. They could increase the production of nitric oxide (NO), along with the activation of macrophages and stimulation of their phagocytic activity [43]. In addition, AGE supports immune system homeostasis, acting in allergic cascade reactions. Kyo and colleagues reported the ability to inhibit the antigen-specific histamine release in the rat basophil cell line RBL-2H3 [44]. AGE treatment of an established allergic-airway inflammation murine model determined a decrease in the main allergic features [45].

## 2. Inflammatory Bowel Diseases: How Can Garlic Be Beneficial?

The anti-inflammatory and immunomodulating properties of AGE continue to arouse the scientific community's interest. Many researchers critically examine the immunoregulation processes mediated by garlic extracts and their active ingredients to find new therapeutic agents that can help treat and prevent diseases such as obesity, metabolic syndrome, cardiovascular disorders, cancer, and inflammatory bowel diseases (IBD). The term IBD includes chronic diseases (such as Ulcerative colitis and Crohn's disease) caused by the inappropriate activation of mucosal immunity and resulting from the interaction between genetic and environmental factors that influence immune responses [46,47].

The discovery of new alternative agents with high efficacy, low toxicity, multi-target, and benefits in easing the complications of IBD is widely requested because the continuous stimulation of the immune system results in the partial destruction of the intestine, causing the main symptoms of the disease. Although the etiology remains largely unknown, recent research has indicated that genetic alterations, the external environment, gut microbial flora, and immune responses are all involved in the pathogenesis of IBD [47,48]. Among the genes that have been associated with an increased risk of IBD, NOD2 and ATG16L1 are involved in the regulation of the immune response. For example, variants of the NOD2 gene are less effective in recognizing and fighting endoluminal germs by triggering inflammatory reactions [48,49]. The malfunction of the ATG16L1 gene, normally involved in the autophagic elimination of intracellular bacteria, culminates in the induction and increase in the expression of pro-inflammatory cytokines such as IL-1$\beta$ [47,49,50]. Numerous studies have also shown that the dynamic balance between the commensal intestinal flora and the host's defensive responses at the mucosal border plays a fundamental role in the pathogenesis of IBD. A typical manifestation of the disease is damage to the intestinal mucosa, which is reflected in defects in the protective mucous layer and loss of tight junctions between epithelial cells [51]. The increased permeability of the epithelial barrier leads to greater access to immunostimulating signals and local immune cells, including dendritic cells (DCs), macrophages, and innate lymphocytes, react by secreting a cocktail of pro-inflammatory cytokines that trigger an inflammatory cycle, which further damages the

intestinal barrier and keeps the inflammatory state active, causing chronic autoimmune inflammation [52].

The conventional approaches to treat IBD include pharmacological therapy (aminosalicylates, corticosteroids, immunomodulators, pro-inflammatory cytokine inhibitors, integrin antagonists, and antibiotics), which aims to reduce inflammation and relieve symptoms. These do not represent decisive treatments but only palliative care since the problem is not eradicated [53]. In addition, many patients may lose response over time. However, it should be emphasized that the primary targets of pharmaceutical treatments are immune system components, and since the substances found in AGE can affect the immune system's homeostasis, they may serve as the basis for the creation of substitute therapies.

Even before the studies on the benefits of AGE, the therapeutic potential of garlic against IBD was already known. For example, extract from fresh garlic was demonstrated to reduce in vitro leukocyte Th1 and inflammatory cytokine production [54]. Allicin, the molecule responsible for the pungent odor of garlic, revealed its ability to modulate nuclear factor-κB activation (NF-κB) activity, suppressing anti-apoptotic and inflammatory target genes in human SW620 and HCT116 colon cancer cells [14]. This effect is of particular interest since patients affected by IBD have an increased risk of developing colon cancer [55].

Another important consideration is that several antibody-induced autoimmune hematologic diseases, such as immune thrombocytopenia (ITP), characterized by platelet destruction by antibodies, have occasionally been observed in patients with IBD [56]. In fact, more and more scientists are conducting studies in order to understand the correlation of these two diseases.

The higher incidence of ITP in childhood rather than in adulthood leads us to move away from the hypothesis that ITP is a consequence of IBD. However, this hypothesis, as demonstrated by Guarina et al. (2021), cannot be excluded since, in pediatric patients, there was complete recovery from ITP after undergoing colectomy [56]. In other patients, a combination of IBD exacerbation and lowering of the platelet count was observed. Furthermore, no therapy for IBD other than steroids appears to have an effect on the course of ITP and the opposite [56]. Therefore, if we consider the benefits of garlic, in particular the immunomodulatory ones, it could be hypothesized that, in those patients affected not only by IBD but also by autoimmune diseases such as ITP, garlic can also exert its therapeutic effect on them [56]. Garlic may play an important role in other blood diseases such as haemophilia A (HA). Currently, this type of hemophilia is treated with blood derivatives, that is, with recombinant factor VIII products (rFVIII) which allow the prolonging of its own half-life [57].

The increase in the purity of these molecules, due to the continuous improvement of their production methods to eradicate the risk of transmission of infectious diseases, has made these drugs more susceptible to the development of alloantibodies (inhibitors) against factor VIII in patients with HA [57].

Therefore, considering the need to minimize the risk of inhibitor development, one can think of the idea of using garlic as an additive to modulate the homeostasis of the immune system in order to reduce the probability that alloantibodies develop [57].

These similar effects can be seen in AGEs as well, which have the benefit of being improved formulations due to the presence of compounds that heighten the positive effect but are absent in fresh preparations, such as SAC, for example. An in vitro study performed by Ide and Lau showed that AGE's most important compound SAC could inhibit Tumor Necrosis Factor-α (TNF-α) and induce NF-κB. Among the main targets of SAC, special attention should be paid to IL-1β. This cytokine is a driver of IBD [58]; it is produced by macrophages, dendritic cells, and monocytes. Along with TNF-α, it can induce the production of other cytokines, such as IL-8, which acts as a neutrophil chemoattractant [59]. The tissue damage and the onset of inflammatory conditions are a direct consequence of a series of cascade events deriving from the increase of IL-1 levels [60]. The blockade of IL-1β was associated with a reduction of inflammation in the murine model of TNF-α independent Ulcerative Colitis [61]. Different studies highlighted the role of SAC in

lowering IL-1β levels [62], suggesting the possibility of using an SAC-based formulation as supportive therapy for IBD. In fact, in another in vitro study, the administration of garlic extracts on stimulated mononuclear cells of whole and peripheral blood lead to a significant decrease in monocyte IL-12, TNF-α, IL-1α, IL-6, IL-8, T-cell interferon-gamma (IFN-γ) and IL-2, and an over-regulation of the production of IL-10, which inhibits the production of anti-inflammatory cytokines and contributes to tissue homeostasis. However, the role of SAC would not end only with the modulation of inflammatory events, but also influencing the intestinal microbiota, whose alterations we know are linked to the onset of IBD. SAC supplementation resulted in improving microbiota homeostasis in colon cancer chemical-induced mice [63] (Figure 3).

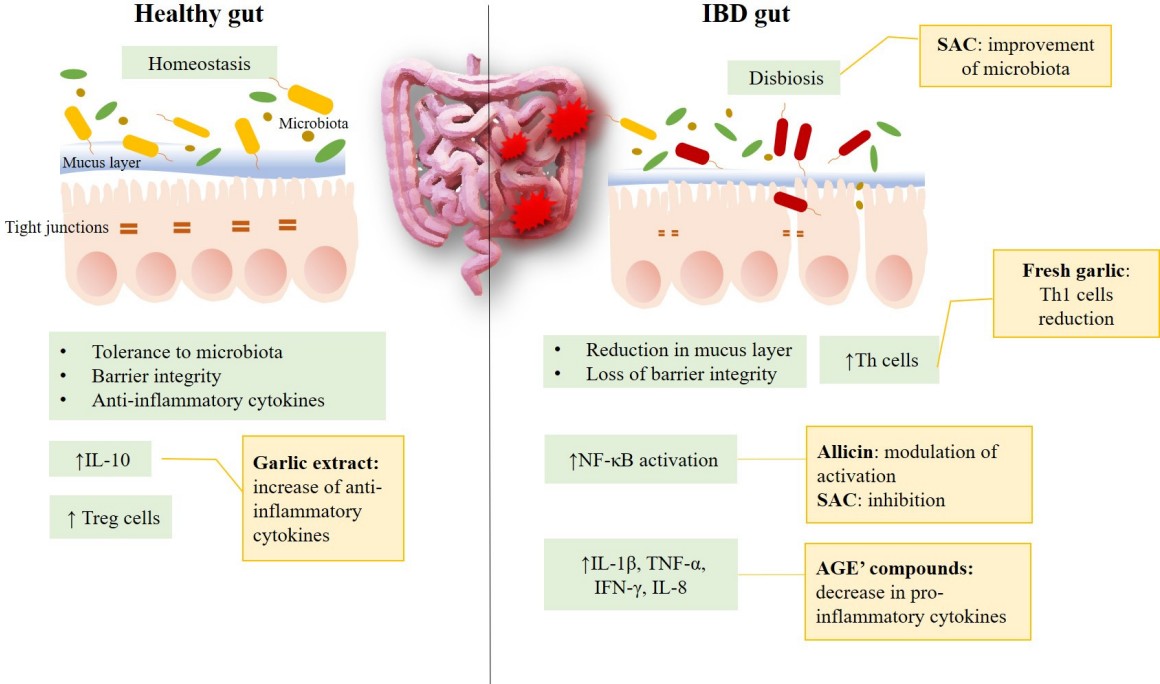

**Figure 3.** Some targets of garlic and its components at the intestinal level. Garlic and its compounds affect intestinal well-being at different levels. In the healthy gut, they can contribute to increasing the anti-inflammatory cytokines; in the case of disease, they have a role in improving microbiota and reducing inflammation, since they act on the key elements of IBD inflammation.

The fact that individuals with IBD have a higher chance of getting colon cancer is one of the reasons why it is crucial to treat inflammation quickly. Given that anti-tumor benefits have been discovered, AGE may be a significant supplement in this context. To better understand the effects of garlic on colon cancer prevention, Shirin et al. (2021), treated two human colon cancer cell lines (SW-480 and HT-29) with S-allylcysteine (SAC) and S-allylmercaptocysteine (SAMC) discovering that SAMC, but not SAC, can inhibit the growth of both cell lines [37]. These effects of SAMC were accompanied by increased Caspase 3 activity and activation of jun kinase showing that increased garlic consumption could lead to a reduced incidence of colon cancer in various human populations [54].

The results observed in numerous in vitro experiments are further corroborated by those found with animal experimentation. For example, Balaha et al. (2016) evaluated the protective effect of garlic oil (GO) by equipping a rat model with induced ulcerative colitis with dextran sulfate sodium (DSS) present in drinking water [64]. The comparison between two groups of this experimental model, after the administration of increasing doses of Mesalazine (MS) and GO, respectively, revealed that GO is able to inhibit DSS-induced colitis in rats thanks to its antioxidant, anti-inflammatory, and immunomodulatory properties. Similar to the therapeutic action of MS, GO suppressed colonic weight gain,

myeloperoxidase (MPO) activity, and significantly reduced malondialdehyde (MDA), TNF-α and IL-1β levels. It concomitantly led to an increase in superoxide dismutase (SOD), glutathione (GSH), and IL-10 levels [64].

These results are in agreement with those shown by Tanrıkulu et al. (2020), which show the ability of GO to reduce colonic inflammation in the acetic acid-induced colitis model in rats [65].

The experimental work conducted by Fasolino et al. (2015) highlighted the anti-inflammatory effect that diallyl sulfide (DAS) and diallyl disulfide (DADS), two organosulfuric compounds of garlic, have on inflammation of the intestinal tract in mice previously subjected to intracolonic administration of dinitrobenzenesulfonic acid (DNBS) [66]. Three days after the induction of colitis, the mice were treated (orally) with DAS and DADS, at doses of 0.3–10 mg/kg. As a consequence of the treatment, a reduction in body weight loss was observed together with a reduction in the weight-to-length ratio of the colon and a remission of mucosal and submucosal edema [66]. Other in vivo studies, in addition to confirming the antioxidant, anti-inflammatory and immunomodulatory properties of garlic, also highlight the ability that the active ingredients of garlic have to modulate the composition of the intestinal microbiome [47,67]. Specifically, an increase in the diversity of probiotics such as Lactobacillus and Bifidobacteria has been observed, which are known to improve intestinal mucosa and intestinal immunity [67].

Furthermore, in mice with DSS-induced colitis, the beneficial effect of OSCs on maintaining the integrity of the intestinal barrier was observed. In the latter, after being fed with roasted garlic, it was possible to observe a significant reduction in intestinal permeability accompanied by an increase in the expression of tight junction proteins and goblet cells compared to the DSS group [68].

## 3. The Development of Alternative Models

The ability to commercialize formulations for the treatment of intestinal disorders is also and foremost dependent on how quickly the studies to establish their efficacy are completed. Generally, two-dimensional models are used in the preliminary assessment. It is common practice to set up an in vitro co-culture model resembling the intestine using Caco-2 intestinal epithelial cells (immortalized cells deriving from human colorectal adenocarcinoma) and THP-1 monocyte cells [69]. For the realization of this model and to imitate the inflamed intestine as closely as possible, THP-1 monocytic cells are plated and treated with phorbol 12-myristate 12-acetate (PMA) and then differentiated into THP-1 macrophage cells. Contextually, Caco-2 cells are cultured and differentiated with IFN-γ on transwell inserts. After the formation of a fully differentiated Caco-2 monolayer, the two cell lines are placed on and made to communicate by inserting the transwell into the plate preloaded with THP-1 cells, usually pre-stimulated with LPS [69]. Then, the evaluation of the anti-inflammatory and immunomodulatory properties of the compounds may be performed, such as the main OSCs deriving from the aged garlic extract, determining the expression levels of the main pro-inflammatory cytokines and evaluating, in addition to their antioxidant power, the integrity of the intestinal barrier.

This co-culture system (Caco-2/immune cells) is one of the most commonly used in the screening of therapeutic molecules due to the ease and speed with which it is possible to study the pharmacological effects of the molecules of interest. Furthermore, the use of Caco-2 cells allows the creation of a system similar to the intestine since these cells maintain similar properties and characteristics to human intestinal tissue, such as permeability and transport of substances [69].

Although 2D cell culture models are commonly used in drug screening projects, the resulting information cannot be considered exhaustive [70]. The incomplete information is attributable to the absence of the surrounding microenvironment, which does not take into account the crosstalk between different types of cells and the inability to reproduce structures similar to crypts and/or villi, nor the degree of complexity of an organ.

These limitations may be overcome through the development of three-dimensional cell cultures, defined as organoids, that can derive from embryonic stem cells, induced pluripotent stem cells, or adult stem cells deriving from the tissue of a certain organ whose use is currently the most practical approach [70]. The use of stem cells allows the reproduction of a variety of organoids, including intestinal organoids. The latter constitute 3D structures capable of at least partially reproducing the identity, behavior, and cellular heterogeneity of the intestinal epithelium in vitro. They, therefore, offer enormous potential for building drug screening platforms and for modeling the course of diseases such as IBD [70,71].

One of the major advantages is the possibility to set up and compare intestinal organoid cultures from healthy subjects with intestinal organoid cultures from IBD patients. In this way, an analysis could be conducted on the biological role played by pro-inflammatory factors in the context of IBD and the therapeutic effect of OSCs could be observed [70,71].

In addition, organoid cultures can be cultivated for long periods without any alterations of the karyotype and phenotype, and cryopreserved without alterations occurring with thawing and freezing [71]. However, in the field of organoid cultures, there are also numerous limitations associated with the need to constantly supply growth factors, the difficulty of reaching the apical surface due to the presence of a closed lumen, and the need to supply oxygen and nutrients by diffusion.

For this reason, many scientists are dedicated to the improvement and development of new more complex technologies, among which organ-on-chip technology is emerging [70,71]. This technology aims at the creation of a small three-dimensional and functional unit that reproduces the exact biochemical microenvironment, the functions, and the same mechanical stresses to which the cells are subjected in our body. At the basis of the functioning of these models is the microfluid through which small quantities of fluids are manipulated within micrometric channels assembled on a single microchip. The latter is made up of constantly perfused hollow chambers, in which living cells are housed and arranged to form the same physiology as the tissue of an organ. Due to the intestinal anatomical complexity and the physiological functions of the intestine, the realization of organ-chip microfluidic models of the human intestine has been a formidable challenge [72].

These models, known as "gut-on-chip", have a microchip to which microchannels are connected and flanked by two hollow chambers which, in addition to allowing a well-controlled laminar flow fluidly, allow the active perfusion of the culture medium, growth, nutrients, as well as of pharmacological compounds. Within this structure, on the upper surface of a porous membrane coated with extracellular matrix (ECM), intestinal epithelial cells are grown and spontaneously acquire the same morphology of villi and crypts observed in the living intestine. On the lower surface of the membrane, the endothelial cells that make up the vascular endothelium are instead grown. Furthermore, white blood cells, commensal microbes, and pathogenic bacteria can also be integrated into this model by reproducing an intestinal microenvironment in which crosstalk occurs between bacterial pathogens, commensal microbes, and immune and vascular cells. All of these characteristics mean that the gut-on-chip can be used for the modeling of human intestinal diseases such as IBD and pharmaceutical studies. For example, it is possible to induce the secretion of pro-inflammatory cytokines with the addition of lipopolysaccharide endotoxin in the luminal microchannel. Therefore, the whole system could be exploited to obtain new knowledge, as well on the mechanisms of inflammatory bowel disease, also on the modulatory and therapeutic effects associated with OSCs of age. Furthermore, the use of organ-on-chip models, in addition to giving the possibility of obtaining a set of information that cannot be obtained with conventional in vitro models, could favor the reduction of animal experimentation. However, the preparation and use of microfluidic systems today constitute a challenge for the future since they require different and complex models of manipulation, the use of distinct culture surfaces, and the optimization of protocols [70,72,73]

## 4. Conclusions

In conclusion, the gastroprotective activity of AGE can therefore be associated with its ability to eliminate oxidative stress and, therefore, with its antioxidant properties with which it not only reduces the levels and activity of malondialdehyde (MDA) and myeloperoxidase (MPO), but restores the activity of antioxidant enzymes such as glutathione peroxidase (tGSH), superoxide dismutase (SOD), and catalase (CAT) to normal. It also acts, on the one hand, by directly enhancing the immune system through the stimulation of the main cells such as lymphocytes, macrophages, natural killer cells (NK), eosinophils and dendritic cells and, on the other hand, modulating the secretion profiles of cytokines by reducing the production of pro-inflammatory cytokines. Therefore, the results of the various experimental studies conducted on this natural product suggest that AGE could be a potential new drug with which to prevent the onset of gastric damage attributable to the intake of NSAIDs, or it could be used in combination with a classic therapy for the treatment of inflammatory bowel diseases to have a greater therapeutic effect. Finally, the development of the appropriate formulations requires the use of modern approaches; in this background, microfluidic technology, represented by organs-on-a-chip, could play an important role in overcoming the limitations given by traditional in vitro or in vivo models.

**Author Contributions:** Conceptualization, E.B.; writing—original draft preparation, S.Z.; writing—review and editing, G.C. and E.B.; figures, G.C and S.Z. All authors have read and agreed to the published version of the manuscript.

**Funding:** This research received no external funding.

**Conflicts of Interest:** The authors declare no conflict of interest.

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
