# Peer review of "Garlic (Allium sativum L.) as an Ally in the Treatment of Inflammatory Bowel Diseases"

_cimb, doi:10.3390/cimb45010046_

Round 1
Reviewer 1 Report
This review article elegantly summarises and discusses most of the knowledge related to garlic extract's beneficial effect, with a particular emphasis on its anti-inflammatory or immunomodulatory functions.
The manuscript is well-written and appealing to the reader. However, in my opinion, section 2 should contain more data regarding the effects of garlic (or its derivatives) in in vivo IBD models (please see PMID: 35992316; 32589234; 26780265; 25488545; 33095019). Since the main objective of the review is to show the relevance of garlic on those chronic inflammatory diseases, results from animal experimentation would help to support the Author´s conclusions.
Author Response
We thank the Reviewer for her/his suggestion that will improve our manuscript. We add the new references PMID: 35992316; 32589234; 26780265; 25488545; 33095019
Reviewer 2 Report
Dear Authors,
I read with interest your review paper on "Garlic (Allium Sativum L.) as an ally in the treatment of Inflammatory Bowel Diseases".
My comments will not significantly affect the content of the paper, but please take note.
First of all, what I see is an error in the binominal name of the garlic species described. The second part of the name should be written with a lower case letter. Perhaps this is too much purism, but it should be so.
Similarly, in Table 1, next to antimicrobial and antifungal properties, you give Latin names as examples - once abbreviated, once full, but all without affiliation (except C. albicans - but wrong). Please standardise this.
I would put polyphenolic ingredients in Fig 1. although "proximate".
In line 308 the manner of citation deviates from the other citations - instead of Shirin et al., (2021) should be modified to include [37].
in line 392, there is no need for a hyphen[62, 64-65], a comma is sufficient.
It seems to me that it might also make sense to cite one of the co-authored papers:
Giulia Caioni, Angelo Viscido, Michele d’Angelo, Gloria Panella, Vanessa Castelli, Carmine Merola, Giuseppe Frieri, Giovanni Latella, Annamaria Cimini and
Elisabetta Benedetti: Inflammatory Bowel Disease: New Insights into the Interplay between Environmental Factors and PPARγ. Int. J. Mol. Sci. 2021, 22(3), 985;
https://doi.org/10.3390/ijms22030985.
Author Response
We thank the Reviewer for her/his suggestions that will improve our manuscript. We correct the binominal name of garlic, and the name of microbes and fungi. We put polyphenolic ingredients in Fig 1. We correct the reference [37] and we put comma at line 392. Finally we add the co-authored papers:Giulia Caioni, Angelo Viscido, Michele d’Angelo, Gloria Panella, Vanessa Castelli, Carmine Merola, Giuseppe Frieri, Giovanni Latella, Annamaria Cimini and Elisabetta Benedetti: Inflammatory Bowel Disease: New Insights into the Interplay between Environmental Factors and PPARγ. Int. J. Mol. Sci. 2021, 22(3), 985;
Reviewer 3 Report
Good work in an hot topic. Please add and comment these references: (doi: 10.3390/jcm10091940 the association with thrombocitopenia and possible speculations) and (doi: 10.3389/fmed.2019.00261 the garlic immunosppressive role in hemophilia inhibitor)
Author Response
We thank the Reviewer for her/his suggestions that will improve our manuscript. We add and comment the references suggested.